# A Double-Blind Crossover RCT Analyzing Technical and Clinical Performance of Monolithic ZrO_2_ Implant Fixed Dental Prostheses (iFDP) in Three Different Digital Workflows

**DOI:** 10.3390/jcm10122661

**Published:** 2021-06-16

**Authors:** Aiste Gintaute, Karin Weber, Nicola U. Zitzmann, Urs Brägger, Marco Ferrari, Tim Joda

**Affiliations:** 1Department of Reconstructive Dentistry, University Center for Dental Medicine Basel, University of Basel, 4058 Basel, Switzerland; aiste.gintaute@unibas.ch (A.G.); n.zitzmann@unibas.ch (N.U.Z.); 2Private Dental Office, 4314 Zeiningen, Switzerland; karinwe17@hotmail.com; 3Department of Reconstructive Dentistry and Gerodontology, School of Dental Medicine, University of Bern, 3010 Bern, Switzerland; urs.braegger@zmk.unibe.ch; 4Department of Prosthodontics and Dental Materials, University of Siena, 53100 Siena, Italy; ferrarm@gmail.com

**Keywords:** dental implant, fixed dental prosthesis (FDP), monolithic, zirconia, zirconium-dioxide (ZrO_2_), digital workflow, accuracy, precision, clinical trial

## Abstract

This double-blind randomized controlled trial with a crossover design analyzed the technical and clinical performance of three-unit monolithic ZrO_2_ implant-fixed dental prostheses (iFDPs), prepared using two complete digital workflows (Test-1, Test-2) and one mixed analog–digital workflow (Control). Each of the 20 study patients received three iFDPs, resulting in 60 restorations for analysis. The quality of the restorations was assessed by analyzing laboratory cross-mounting and calculating the chairside adjustment time required during fitting. All iFDPs could be produced successfully with all three workflows. The highest cross-mounting success rate was observed for the original pairing iFDP/model of the Control group. Overall, 60% of iFDPs prepared with Test-1 workflow did not require chairside adjustment compared with 50% for Test-2 and 30% for Controls. The mean total chairside adjustment time, as the sum of interproximal, pontic, and occlusal corrections was 2.59 ± 2.51 min (Control), 2.88 ± 2.86 min (Test-1), and 3.87 ± 3.02 min (Test-2). All tested workflows were feasible for treatment with iFDPs in posterior sites on a soft tissue level type implant system. For clinical routine, it has to be considered that chairside adjustments may be necessary, at least in every second patient, independent on the workflow used.

## 1. Introduction

Digitization has significantly influenced dentistry in recent years and continues to enable new options in clinical routines [1]. Translating established, conventional dental procedures into digitized protocols requires a detailed understanding of digital dental processes [2]. Conventional procedures cannot always be transferred identically to digital workflows. Continuous progress inevitably leads to an adaptation of the established workflows in the clinic and in dental technology. Consequently, the digital processes must be critically analyzed and recalibrated [3].

Intraoral optical scanning (IOS) plays a crucial role in digital dentistry, especially in the field of prosthodontics [4,5]. The contactless transfer of the individual patient situation to the virtual dental laboratory, without the use of any physical models, is central to completely digital workflows [6]. The continuous and incremental improvement in the quality of IOS systems, both hardware and software, is enabling the production of more and more accurate and precise final restorations [7]. In this context, the level of digital experience of the dental team (operating the different IOS and CAD/CAM systems) has a significant impact on the quality of the final outcome [8]. While the interfaces of available digital systems have been opened, enabling free data transfer between systems from different manufacturers, it is still unclear whether there is a loss of data quality and, subsequently, a fluctuating quality of reconstructions when non-proprietary data is transferred between different CAD/CAM systems [9].

Among fixed dental prostheses (FDP) supported by dental implants, digital pathways have demonstrated clear superiority compared with conventional workflows for the production of single implant crowns. Implant crowns fabricated digitally were superior in terms of clinical fitting [10] and economic parameters, such as time-efficiency [11] and costs [12] compared with classical impressions and gypsum casts. The use of standardized scan bodies makes the IOS technology predestined for therapy with fixed implant reconstructions. Whereas in single restorations, secure occlusion is usually ensured by the adjacent teeth, in multi-unit implant-supported fixed dental prostheses (iFDP), the lack of stable occlusal support poses the challenge for IOS systems during computerized bite registration [13]. It is not the optical resolution of the IOS devices or the power of the software that are the limiting features, rather that the missing occlusal units become the bottleneck in digital impression taking for iFDPs [14]. The question arises to what extent dental models are required for fabricating iFDPs, or whether a completely digital approach is possible, similar to what can be done for single implant crowns, i.e., IOS > Design and Mill > Delivery [15].

While the dental technology industry may promise that their IOS systems will deliver smooth processes with reproducible results in prosthodontic indications, clinical evidence is lacking. In particular, evidence for the feasibility of fully digital workflows for iFDPs, starting with IOS in the clinic, to CAD/CAM-processing in the dental lab, and back to clinical delivery. Therefore, the aim of the present study was to analyze the technical and clinical performance of monolithic zirconium-dioxide (ZrO_2_) iFDPs prepared according to two complete digital workflows and one mixed analog–digital workflow. The null hypothesis was that the process quality of iFDPs in these three digital workflows is similar for the two investigated outcomes, namely (i) laboratory cross-mounting; and (ii) clinical performance indicator (CPI) defined by restoration fit, chairside adjustment time, and consecutive density changes of the modified iFDPs.

## 2. Materials and Methods

### 2.1. Trial Setting

The study was designed as a prospective, double-blind, triple-armed randomized controlled trial (RCT) with a crossover approach in a university-based setting. Neither the clinician nor the patient was aware of the three different treatment groups. The protocol was officially approved by the Ethics Committee Basel, Switzerland (EKNZ-ID 2019-00706) and registered at ClinTrials.gov (NCT 04029025). This RCT was conducted in compliance with the study protocol, the current version of the Declaration of Helsinki, the ICH-GCP, as well as all national legal and regulatory requirements. Patients provided an informed consent to participate in the trial. No changes were made to methods after trial commencement. The RCT followed the CONSORT 2010 statements (http://www.consort-statement.org/consort-2010, accessed on 19 April 2021).

At the time of development of the study protocol, no data from trials investigating the clinical and technical performance of three-unit monolithic ZrO_2_ iFDPs were available. Therefore, the power analysis for the present study was based on our own preliminary findings for treatment with implant-supported single units considering time efficiency in terms of clinical adjustment time: with a coefficient of determination of R^2^ = 0.5, a statistical power of 0.8, and a significance level of α = 0.05, a sample size of *n* = 16 would be required. Therefore, 20 study patients presenting with a three-unit edentulous space or free-end situation in one posterior quadrant qualifying for a three-unit FDP supported by two dental implants (Tissue Level Implant System RN/WN, Institut Straumann AG, Basel, Switzerland) were considered sufficient. Inclusion criteria were periodontal health or successfully treated [16], consumption of no more than 10 cigarettes per day, the presence of antagonistic contacts, and at least one adjacent tooth.

Each of the 20 study patients received three iFDPs prepared, according to three different workflows, resulting in a total of 60 restorations. All iFDPs were designed and produced out of monolithic ZrO_2_ (VITA YZ ST Super Translucent Multicolor, Bad Säckingen, Germany) as three-unit screw-retained, full-contour reconstructions (Figure 1). Two of the workflows were completely digital and the third combined analog and digital steps:**Test-1 “Complete Digital Workflow”** (3Shape, Copenhagen, Denmark) IOS Trios 3 + Dental System Lab-Software;**Test-2 “Complete Digital Workflow”** (Dental Wings Inc., Montreal, Canada) IOS Virtuo Vivo + DWOS Lab-Software;**Control “Analog–Digital Workflow”** Polyether Impression/Gypsum Cast/Lab-Scan + EXOCAD Lab-Software.

The baseline of the study was the start of the prosthetic therapy. The study involved three key stages: (i) impression-taking for clinical registration of the implants’ 3D-positioning, including antagonists and occlusal relation; (ii) CAD/CAM fabrication of iFDPs; and (iii) clinical try-in/delivery (Figure 2). Distribution of the work steps, whether beginning with Test-1, Test-2, or Control, as well as the order of sequence during iFDP try-in, were randomly chosen per study patient using the envelope-technique. The principal investigator (T.J.) performed the random allocation sequence and the enrollment of all study patients. Blinding persisted at the time of try-in/delivery of the final iFDPs to the study patients and to the clinical operator. For each of the three workflows, outcomes were evaluated according to performance of the produced reconstructions based on (i) the feasibility of laboratory cross-mounting of each iFDP; and (ii) clinical fit and assessment of adaptation time for clinical adjustments of interproximal surfaces, pontic areas and occlusal surfaces—if required.

### 2.2. Clinical and Technical Work Steps

Clinical work steps were performed by one experienced dentist (K.W.) and observed by one spectator (A.G.). For the complete digital workflows (Test-1, Test-2), the intraorally obtained STL-files were directly transferred to the relevant laboratory CAD/CAM software specific for the IOS system used: Trios 3 > Dental System Lab-Software (3Shape, Copenhagen, Denmark); and Virtuo Vivo > DWOS Lab-Software (Dental Wings Inc., Montreal, Canada). Full-arch digital impressions were taken and scanned according to the manufacturer’s recommendations.

For the analog–digital workflow (Control), the pick-up impression technique with individualized open trays and polyether impression material (Impregum, 3M ESPE, Neuss, Germany) was used. In addition, a high-viscosity alginate impression was taken from the antagonistic arch (Palgat Plus Quick, 3M Espe GmbH, Neuss, Germany) as well as occlusal registration with fast-setting vinyl polysiloxane Blu-Mousse (Parkell Inc., Edgewood, USA). Gypsum implant master casts were manmade, conventionally, with low-expansion die stone ISO Type 4 (Silky Rock, Whip Mix, Louisville, KY, USA) under consideration of the company’s recommendations. The gypsum implant master casts were scanned in the laboratory (Ceramill Map 400+, Amann Girrbach, Koblach, Austria) and the STL-files further processed with EXOCAD Lab-Software (EXOCAD, Darmstadt, Germany).

Finally, all 60 monolithic ZrO_2_ iFDPs were milled with a five-axis unit (Ceramill 2 Motion, Amann Girrbach, Koblach, Austria) and bonded to pre-fabricated titanium base abutments (Variobase RN/WN, Institut Straumann AG, Basel, Switzerland). For both complete digital workflows, the iFDPs were finalized without using physical models.

### 2.3. Laboratory Cross-Mounting

Only for testing cross-mounting of the restorations, implant models for Test-1 and Test-2 were additionally made out of a high-precision photopolymer dental resin (Form 3B, Formlabs Inc., Somerville, MA, USA) matched to the 3D printer system (Model Resin 1L, Formlabs Inc., Somerville, USA). For each study patient, iFDPs of one group were placed onto corresponding model situations of the other groups and vice versa. Success was defined as a fit to the implant analogues, interproximal plus occlusal fit in terms of dichotomic feasibility testing for all parameters (yes/no).

Calibration among the evaluators was completed in advance of the RCT. Evaluators were trained using pairs of models/reconstructions that represented mis-fitting and well-fitting situations. Analysis was done by the clinical operator (K.W.) and independently verified by the observer (A.G.). Evaluation was performed separately by the clinical operator and the observer including repetition after one week with an inter-examiner Kappa-Score of 1.0 (Figure 3).

### 2.4. Clinical Performance Indicators (CPI)

The CPIs comprised the clinical fit of the iFDPs, the chairside adjustment time required to fit the iFDPs, and the subsequent density changes of the adjusted iFDPs (if applicable). For clinical assessment, the healing abutments were removed and the iFDPs were mounted onto the implants with respect to the randomization process for the order of the sequence and respecting the prerequisite of blinding. Only the dental technician and the clinical observer were informed about the traceability from the iFDPs and their related workflows.

First, the interproximal fit and the seating of the iFDPs were analyzed, striving for continuity with waxed dental floss at neighboring sites. If necessary, corrections were made with diamond burs and silicone polishers to create satisfactory interproximal surfaces. Secondary, the pontic site was evaluated, aiming at Superfloss (Oral-B, Procter Gamble, Cincinnati, OH, USA) passing through with some hindrance. Finally, the occlusal scheme was checked, aiming at occlusal contacts with 12 μm articulation foil in maximum intercuspation without contacts during mandibular movements. Again, if necessary, adjustments were made as described to achieve light occlusal contacts without dynamic interference. The observer documented the treatment time needed for each clinical step to ensure that the treatment time was accurately recorded.

Afterwards, all iFDPs in need of adjustments were analyzed for deviation compared to the original design. The clinically modified iFDPs were digitized by the same lab-side scanner, which was used previously for scanning of the gypsum implant master casts the Controls. STL-files were then imported into a 3D analysis software (PreForm, Formlabs Inc., Somerville, MA, USA) and matched with the original virtual reconstructive design related to the workflows of Test-1, Test-2, and Control. A best-fit algorithm of the 3D analysis software was applied for volumetric deviation analysis in terms of density changes in g/mm^3^ and visualization of the superimposed STL-files (Figure 4).

### 2.5. Statistical Analysis

Statistical analysis was conducted with the program “Software R” (version 4.0.4). Kruskal–Wallis Tests were used for all comparisons in this RCT with crossover design. Since no carry-over effects were expected, the data of investigated measurement-rounds were used for analysis of Test-1, Test-2, and Control, respectively. A *p*-value of < 0.05 was considered as statistically significant.

## 3. Results

The mean age of the patients was 63 years (range 30 to 76 years) and 55% were female. The location of the iFDPs were equally distributed between the maxilla and mandible, and most implants had a regular neck configuration (*n* = 35 RN and *n* = 5 WN). A total of 17 iFDPs were examined in free-end situations and three iFDPs in edentulous spaces with two adjacent teeth. All analyses were performed on originally assigned groups. Recruitment started in January 2020 and no study patient was excluded after enrollment.

### 3.1. Laboratory Cross-Mounting

Based on the three workflows, all 60 iFDPs were tried in onto the conventionally manufactured gypsum implant casts (Control) and onto the 3D printed implant models (Test-1 and Test-2), resulting in nine possible pairings for laboratory cross-mounting (Table 1). Correlation analysis of crossover investigation revealed that none of the iFDP could be successfully mounted on all three corresponding model situations. The highest transfer success rate was the original pairing iFDP Control/model Control; and the lowest was iFDP Control/model Test-2. Per definition, failure modes of laboratory cross-mounting were characterized by interproximal, and subsequently, occlusal misfit of the iFDPs.

### 3.2. Clinical Performance Indicators (CPI)

All iFDPs could be produced successfully. For Test-1, IOS had to be redone for one study patient due to an unusable STL-dataset. The need for clinical adjustment was highest for iFDPs of Controls, followed by Test-2 and Test-1. Overall, 14 iFDPs of the mixed analog–digital workflow required clinical corrections to achieve sufficient fit in terms of interproximal contact and occlusion; whereby 10 (Test-2) and 12 (Test-1) iFDPs of the complete digital workflows could be inserted without any modifications, representing success rates for adjustment-free delivery of 30% (Control), 50% (Test-2), and 60% (Test-1), respectively. In three (15%) of the 20 study patients, all three iFDPs could be immediately inserted without any corrections. Results for the mean total adjustment time, as the sum of interproximal plus occlusal corrections plus pontic-area modifications, were 2.59 min (SD ± 2.51) for Control, 2.88 min (SD ± 2.86) for Test-1, and 3.87 min (SD ± 3.02) for Test-2. The mean values for density changes of the clinically modified iFDPs were 4.92 g/mm^3^ (SD ± 9.64) for Test-1, 35.83 g/mm^3^ (SD ± 114.14) for Test-2, and 14.13 g/mm^3^ (SD ± 40.51) for Control.

## 4. Discussion

This double-blind, crossover RCT investigated digital and conventional workflows for producing monolithic ZrO_2_ iFDPs in posterior sites. The results revealed differences in the technical and clinical performance of the three different treatment workflows. Therefore, the tested hypothesis for equivalence for the two complete digital workflows plus one mixed analog–digital workflow had to be rejected.

Oral rehabilitation of partially edentulous patients with implant-retained multi-span fixed reconstructions is a comprehensive treatment requiring multiple work steps in an interdisciplinary approach with knowledge and skills in prosthodontics, implant surgery, and dental technology [17]. The team of clinicians and technicians has to rely on the therapy protocols to achieve long-term success. For the implementation of new workflows in clinical routine use, these must withstand the comparison with the established approach [18]. In general, the feasibility of a new therapy’s protocol is the minimal prerequisite; predictable and reproducible treatment outcomes are the key to become a serious alternative to the gold standard–or even to be disruptive enough to replace it [19]. The specific trial setting of the current study with a crossover design allowed direct comparisons of three different workflows for each study patient.

In the present study, laboratory cross-mounting was conducted to provide a simple quality-check of the three iFDP workflows prior to the clinical try-in. Since the three iFDPs derived from the different workflows for each patient were fabricated for the same clinical situation, it should be expected that the implant reconstructions could be exchanged among each other on the corresponding dental models for Test-1, Test-2, and Control, respectively. However, no iFDP could be successfully mounted on all three-model pairings indicating that either the impression technique, the CAD/CAM process, the model fabrication, or a combination of these factors affected this outcome. The completely digitally produced iFDPs had worse laboratory transfer rates than the analog–digital workflow, with Test-1 workflow performing slightly better than Test-2. In this context, it has to be emphasized that iFDPs from the Test-1 and Test-2 workflows were virtually designed and produced without the 3D printed models. These models were used only for cross-mounting analysis. No other investigation has reported on cross-mounting of multi-unit iFDPs based on a prospective clinical trial; however, the present findings are consistent with a previously published investigation of in vitro cross-mounting comparing implant-supported single crowns produced with digital and conventional workflows [10].

In the present study, CPIs comprised clinical fit, the time needed for chairside adjustments, and the subsequent density changes of the adjusted iFDPs. Both the dentist and the patient were blinded to the workflow that was used to generate the individual iFDPs to minimize bias. While the success rate for adjustment-free delivery of iFDPs of Test-1 was minimally better than that of Test-2, the success rate of Test-1 was twice as high as that of Controls. The shortest chair-time was required for correction of Control iFDPs (relevant for cases requiring corrections); however, the overall clinical adjustment time for Test-1, Test-2, and Controls differed only slightly within a narrow range of 2.59 min to 3.87 min without statistical significance. Therefore, the general need for clinical adaptation of iFDPs (yes/no) must be rated higher than the amount of time required for adjustment. For treatment with single-unit implant crowns in posterior sites, clinical trials also reported on complete digital workflows in favor of conventional workflows in terms of time-efficiency [20,21].

The number of adjustments required in the present study was quantified by comparing the initial iFDP design with the modified restoration, and the volumetric deviation was expressed as density changes. Applying this novel method, Test-1 performed more than seven-fold better than Test-2, and almost three-fold better than Controls. Adjustment time and density changes can only represent surrogate parameters as CPIs for the evaluation of the process quality. The best performance indicator, however, is the successful insertion of the iFDP–ideally without corrections to the prosthetic reconstruction [22]. Not only do clinical step-by-step modifications require valuable chairside time, the adjusted surfaces of the iFDPs represent predilection areas and must be re-polished to a high gloss prior to final insertion [23,24].

Overall, the results of the present RCT investigating three-unit monolithic ZrO_2_ iFDPs revealed lower performance with complete digital treatment protocols compared with previously published data for single-unit implant crowns [15]. One possible explanation for this may be that multi-span implant-retained reconstructions lack unilaterally antagonistic pairs for a secure positional relationship of the jaws to each other, depending on the individual patient situation. Based on the present findings, it can only be speculated whether the reason for the need for clinical modifications of the iFDPs is the technical process itself or the patient-specific situation affecting the bite registration [13]. The typical indication for an iFDP is the rehabilitation of posterior free-end situations to reestablish masticatory function without the need for removable dental prostheses. Therefore, it is not surprising that 85% of the iFDPs in the present study were located in free-end situations. This classic treatment “premolar-pontic-molar”, unlike single-unit implant crowns, makes bite registration more complex and less predictable. In the complete digital workflows, the systems work with the data recorded in a single moment, whereas in the mixed analog–digital workflow, the dental technician can incorporate experience in designing the occlusion in the physical articulator. Therefore, digital bite registration still seems to be the critical point today [14].

To interpret the findings from the digital and complete digital workflows in the present study, the specific systems used should be considered. Test-1 and Test-2 exemplified synchronized workflows of IOS and corresponding laboratory CAD-software from the same company, while the Control workflow used lab-side scanning of the dental implant master casts and EXOCAD as laboratory CAD-software. For further production following the CAM-process, all three workflows used the same milling unit and the same ZrO_2_ blanks in order to harmonize the final production and to focus on the initial work steps of data generation and technical processing. In this RCT, only proprietary data transfer within system-specific solutions of STL generation and processing was analyzed. Therefore, future research should also consider whether different data flows involving mixing systems might create better clinical results. Moreover, prospective follow-up studies need to investigate the aspect of digital bite registration in order to define a system-specific minimal number of antagonistic units as threshold indicating if a complete digital workflow is feasible.

Based on the results of laboratory cross-mounting and clinical adjustment time from the present study it can be concluded that all three workflows worked successfully. Conventional impression-taking plus lab-side scanning of dental gypsum casts demonstrated high precision for analysis of the iFDP/model pairing, but not necessarily in the clinical situation. Conversely, the complete digital workflows with IOS and model-free processing achieved higher rates for adjustment-free delivery of the iFDPs in the clinic, while fitting on the 3D printed models fabricated from the STL data was worse. It cannot be generalized that digital workflows using different IOS and CAD/CAM system might produce identical results per se. The important factors seem to be: (i) what system is used for which indication? [25] (ii) Is a proprietary data transfer used? (iii) Are the hardware and software up to date [7]?

Complete digital workflows represent the future of (fixed) prosthodontics. Reducing the absolute number of laboratory steps has the advantage of lower susceptibility to technical errors. Therefore, the potential advantage of digital workflows should not be diminished by leaving the pathway of model-free fabrication. It is feasible to use all three digital workflows for treatment with three-unit monolithic ZrO_2_ iFDPs in posterior sites on a soft tissue level type implant system (with given emergence profile). For a clinical routine, chairside adjustments may be necessary, at least in every second patient, independent on the workflow used.

## Figures and Tables

**Figure 1 jcm-10-02661-f001:**
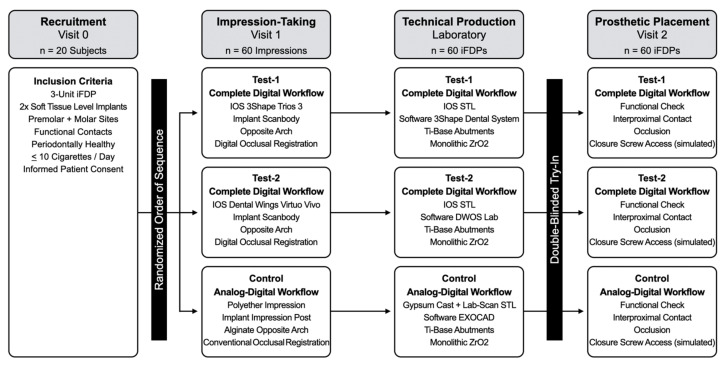
Study flow-chart for clinical and technical work steps.

**Figure 2 jcm-10-02661-f002:**
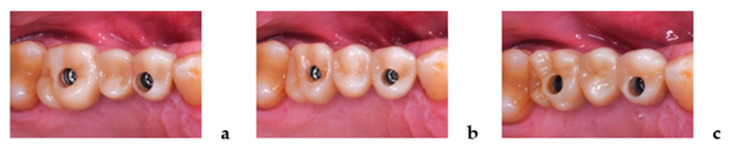
Clinical try-in/delivery of the iFDPs for Test-1 (**a**), Test-2 (**b**), and Control (**c**).

**Figure 3 jcm-10-02661-f003:**
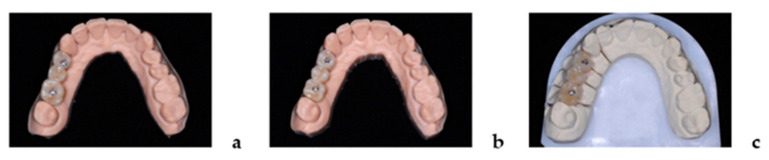
Set-up for cross-mounting displaying dental model situations with inserted iFDPs representing on study patient with 3D-printed implant models for Test-1 (**a**) and Test-2 (**b**), as well as gypsum implant master cast for Control (**c**).

**Figure 4 jcm-10-02661-f004:**

Deviation analysis comparing the initial virtual design and the digitized iFDPs after clinical adjustments for Test-1 (**a**), Test-2 (**b**), and Control (**c**) mounted on the gypsum implant cast guaranteeing a standardized evaluation.

**Table 1 jcm-10-02661-t001:** Feasibility testing (yes/no) for laboratory cross-mounting summarizing success rates of 3 × 20 iFDPs and 3 × 20 model situations in nine possible pairings.

			Model Situation	
		Test-1	Test-2	Control
	**Test-1**	12/20 (60%)	6/20 (30%)	13/20 (65%)
**iFDP**	**Test-2**	3/20 (15%)	5/20 (25%)	12/20 (60%)
	**Control**	3/20 (15%)	2/20 (10%)	15/20 (75%)

## Data Availability

The data are not publicly available due to privacy restrictions.

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
