# Peer review of "A Double-Blind Crossover RCT Analyzing Technical and Clinical Performance of Monolithic ZrO_2_ Implant Fixed Dental Prostheses (iFDP) in Three Different Digital Workflows"

_jcm, 2021, doi:10.3390/jcm10122661_

Round 1

Reviewer 1 Report

I congratulate the authors for complying with most of the essential aspects of a high-quality RCT. However, I recommend that the authors review the following points: 

  1. In accordance with CONSORT guidelines, we recommend that authors explicitly mention the primary outcome on which the sample size calculation is performed and on which the research is focused. The primary outcome should refer to an unique outcome measurement, e.g., clinical adjustment time; all other outcome measurements would be considered secondary outcomes.   2. The authors have described the use of the Wilcoxon rank sum test for all comparisons. However, the Wilcoxon rank sum test is commonly used for comparison of two nonparametric data sets. In this three-arm RCT, the authors should either explain why they did not use the Kruskal-Wallis test, since it is the nonparametric test used to compare three or more sets of sample data, or please explicitly explain how the Wilcoxon rank sum test has been applied between the 3 groups.   3. It is suggested that the authors describe which random component was used in the sequence generation process, for example: computer-generated random numbers; random number table; coin toss; card shuffling.

Author Response

Reviewer: 1

Comments to the Author

I congratulate the authors for complying with most of the essential aspects of a high-quality RCT.

Response

Thank you very much for your time to review our manuscript and for your help in improving the quality of the manuscript.

However, I recommend that the authors review the following points:

  1. In accordance with CONSORT guidelines, we recommend that authors explicitly mention the primary outcome on which the sample size calculation is performed and on which the research is focused. The primary outcome should refer to an unique outcome measurement, e.g., clinical adjustment time; all other outcome measurements would be considered secondary outcomes.

Response

Thanks for your valuable comment. We revised the manuscript according to your recommendation and made it clear that the Sample Size Calculation refers to time-efficiency in terms of clinical adjustment time (text change: lines 93/94).

  1. The authors have described the use of the Wilcoxon rank sum test for all comparisons. However, the Wilcoxon rank sum test is commonly used for comparison of two nonparametric data sets. In this three-arm RCT, the authors should either explain why they did not use the Kruskal-Wallis test, since it is the nonparametric test used to compare three or more sets of sample data, or please explicitly explain how the Wilcoxon rank sum test has been applied between the 3 groups.

Response

That's a very good point – thank you very much. We had initially discussed this issue with our statistical consultant as well. The Wilcoxon Rank Sum Test was applied in pairs (Test-1 vs. Test-2; Test-1 vs. Control; Test-2 vs. Control). However, due to the reviewer's question about the statistical model, we again performed a calculation with our statistician using the Kruskal-Wallis Test as a nonparametric test to compare Test-1, Test-2, and Control. The results are the same. Nevertheless, we have decided (not least because of the reviewer's recommendation) to update the statistical method in the manuscript accordingly (text change: line 203).

  1. It is suggested that the authors describe which random component was used in the sequence generation process, for example: computer-generated random numbers; random number table; coin toss; card shuffling.

Response

Thanks again. We added an explanatory comment describing in detail the random process (text change: lines 118-123).

Reviewer 2 Report

The paper " A double-blind cross-over RCT analyzing technical and clinical performance of monolithic ZrO2 implant fixed dental prostheses in three different digital workflows" is quite interesting and well conducted. The abstract is clear; the introduction is clear and comprehensive. Materials and methods and results are well described. In my opinion, the discussion, properly conducted, is too long and should be summarized .

Author Response

Reviewer: 2

Comments to the Author

The paper "A double-blind cross-over RCT analyzing technical and clinical performance of monolithic ZrO2 implant fixed dental prostheses in three different digital workflows" is quite interesting and well conducted. The abstract is clear; the introduction is clear and comprehensive. Materials and methods and results are well described. In my opinion, the discussion, properly conducted, is too long and should be summarized.

Response

Thank you very much for your time to review our manuscript and for your help in improving the quality of the manuscript.

We are pleased that the reviewer considered the individual sections of the manuscript to be well conducted. We absolutely agree with the reviewer that the length of the discussion is at the upper limit, but we also believe that the content discussed here is important for the overall understanding. Moreover, we followed the Journal’s recommendation to include the Conclusion of the RCT at the end of the Discussion. This also makes up part of the discussion in terms of quantity. Overall, we would like to keep the discussion in the current version.